# Efficacy and Safety of Ginger on the Side Effects of Chemotherapy in Breast Cancer Patients: Systematic Review and Meta-Analysis

**DOI:** 10.3390/ijms231911267

**Published:** 2022-09-24

**Authors:** Soo-Dam Kim, Eun-Bin Kwag, Ming-Xiao Yang, Hwa-Seung Yoo

**Affiliations:** 1College of Korean Medicine, Daejeon University, Daejeon 34520, Korea; 2Integrative Medicine Service, Memorial Sloan Kettering Cancer Center, Department of Medicine, 321 E 61st Street, New York, NY 10065, USA; 3East West Cancer Center, Daejeon Korean Medicine Hospital, Daejeon University, Daejeon 35235, Korea; 4East West Cancer Center, Seoul Korean Medicine Hospital, Daejeon University, Seoul 05836, Korea

**Keywords:** ginger, zingiber officinale, breast cancer, chemotherapy, side-effect, chemotherapy-induced nausea and vomiting

## Abstract

Cancer is one of the leading causes of death in the world, with breast cancer being the most prevalent cancer. Chemotherapy-induced nausea and vomiting (CINV) is one of the most serious side effects of chemotherapy. Because the current CINV treatment option has several flaws, alternative treatment options are required. Ginger has traditionally been used to treat nausea and vomiting, and it also has anticancer properties in breast cancer cells. Based on these findings, researchers investigated whether using ginger to treat CINV in breast cancer patients is both effective and safe. We searched PubMed, Embase, Cochrane Library, CNKI, and Wanfang from inception to June 2022. Outcomes included Rhodes Index Scores of Nausea, Vomiting, and Retching, severity and frequency of CINV. Five RCTs were included. We pooled all included data and performed subgroup analysis by types of CINV. Overall, authors found that ginger was associated with a reduction in CINV. Subgroup and sensitivity analysis revealed that managing severity of acute CINV in breast cancer patients with ginger was efficient. In terms of managing delayed CINV in breast cancer patients, ginger was also statistically significant. The authors concluded that ginger may be helpful in lowering both acute and delayed CINV in breast cancer patients. Since there were no serious side effects, ginger is thought to be safe.

## 1. Introduction

Cancer is one of the leading causes of death in the world. According to cancer statistics in 2022, there are expected to be 1,918,030 new cancer cases and 609,360 cancer-related deaths [1]. Among them, breast cancer has the highest estimated new cases of 287,850 patients (31%) [1]. Chemotherapy, like other types of cancer treatment, is widely used in the treatment of breast cancer. Even though it is a necessary conventional treatment for cancer patients, it has a few serious side effects that can be fatal to cancer patients both during and after treatment and chemotherapy-induced nausea and vomiting (CINV) is one of them [2]. Chemotherapy for cancer patients is divided into four categories: high emetic risk, moderate emetic risk, low emetic risk, and minimal emetic risk [3]. Specifically, in comparison to cisplatin, melphalan, cyclophosphamide, and dacarbazine, which have a high emetogenic potential, anthracyclines, methotrexate, oxaliplatin, and carboplatin have a moderate emetogenic potential [3].

CINV is generally classified into five categories, which are acute, delayed, anticipatory, breakthrough, and refractory [4]. Acute CINV develops within 24 h of starting chemotherapy. Delayed CINV develops after 24 h of chemotherapy. Anticipatory CINV is a general response to chemotherapy. Despite appropriate prophylaxis, breakthrough CINV occurs within 5 days of chemotherapy, and refractory CINV occurs in subsequent chemotherapy cycles after the occurrence of breakthrough CINV in prior cycles, excluding anticipatory CINV [4]. Treatment agents for CINV and a combination of the agents are categorized as minimal, low, moderate, or high and the prevention and treatment strategies are varied depending on the severity of CINV [3]. Different types of treatments include dexamethasone which is a first-line use in combination with other agents [4]. However, side effects including insomnia, indigestion/epigastric discomfort, agitation, increased appetite, weight gain, and acne were reported [5]. Untreated CINV is linked to treatment discontinuation, decreased quality of life, complications such as dehydration and electrolyte imbalances, and, ultimately, decreased treatment success and increased costs of care [5]. Therefore, efforts are being made to establish effective evidence-based clinical guidelines; for example, the ASCO (American Society for Clinical Oncology) has recognized acupuncture as an alternative therapy for CINV and is expected to provide additional alternative therapies as qualified evidence accumulates [6].

Ginger (Zingiber officinale) is a commonly used herb to treat nausea and vomiting, and several bioactive compounds, including shogaols, gingerols, zingerone, and paradols, have been identified within the ginger rhizome [7]. These compounds are thought to interact with a variety of areas involved in the development of CINV [8]. In cases of acute CINV, free radicals produced by toxic chemotherapy drugs stimulate enterochromaffin cells in the gastrointestinal tract, resulting in the production of serotonin [9,10]. Serotonin then binds to intestinal vagal afferent nerves via 5-HT_3_ receptors, causing the vomiting reflex to be triggered in the CNS via the nucleus of the solitary tract (NTS) and chemoreceptor trigger zone (CTZ) [9,10]. Moreover, 5-HT_3_ receptor signaling may be involved in delayed CINV, but to a lesser extent than in acute CINV [10]. Substance P is thought to be the main neurotransmitter involved in delayed CINV [11]. Chemotherapy drugs cause neurons in the central and peripheral nervous systems to release substance P, which then binds to neurokinin-1 (NK1) receptors, primarily in the NTS, to cause vomiting [10].

Ginger’s bioactivity has been studied to see how it affects the CINV mechanism. Ymahara was the first to demonstrate that the whole ginger including gingerols 6,8 and 10 can inhibit 5-HT_3_-induced contraction [12]. Followed by that, the 5-HT_3_ antagonistic effect of ginger using four major compounds of ginger (gingerol 6,8,10 and 6-shogaol) was found in animal experiments [13]. Clinical trials for humans also reported that daily addition of ginger with conventional chemotherapy resulted in a favorable effect on CINV with no adverse effects [14,15,16].

Using ginger for breast cancer patients not only to treat CINV but also to manage the disease has numerous benefits. According to a study conducted in 2017 by Martin and his colleagues, 10-gingerol significantly inhibited metastasis of TNBC (Triple-negative breast cancer) cells dose-dependently [17]. Another investigation found that ginger therapy reduced colony formation and proliferation in MCF-7 and MDA-MB-231 breast cancer cell lines [18]. The non-tumorigenic normal mammary epithelial cell line (MCF-10A) was not severely affected by it, but there was a loss of cell viability, chromatin condensation, DNA fragmentation, activation of caspase 3, and cleavage of poly (ADP-ribose) polymerase. At the molecular level, the upregulation of the Bax and the downregulation of the Bcl-2 proteins may contribute to the apoptotic cell death caused by ginger [19].

Given these overlapping benefits of using ginger in breast cancer treatment, the authors wanted to systematically examine the efficacy and safety of using ginger with CINV as the primary endpoint and others as secondary endpoints. Despite a recent study demonstrating that ginger use in CINV is beneficial [20], since most of the analysis looked at several cancer types, systematic and meta-analyses on breast cancer and CINV are limited [21,22]. Thus, the efficacy and safety of using ginger to manage the side effects of breast cancer treatment, as well as its potential benefits, will be examined and discussed in this study.

## 2. Results

### 2.1. Literature Search Results

We discovered a total of 165 records using search databases. We screened 115 remaining studies for eligible title and abstract after removing duplicates (*n* = 50), and 110 were excluded due to ineligibility or irrelevance (Figure 1). The abstract and title were read, and 47 articles were eliminated. After determining eligibility, 63 articles were excluded because 27 were “Review articles,” 3 were “Not human,” 5 were “Not intervention,” 4 were “Not population,” 9 were “Not study design,” and 15 were “Not full text.” As a result, we finally included five trials in this re-evaluation [23,24,25,26,27].

### 2.2. Characteristics of Included Studies

This review included five randomized controlled trials, and the key characteristics of the included studies are listed in Table 1. The participants in the studies ranged in age from 41.8 to 52.1 years. Sample sizes ranged between 60 and 119 patients. The chemotherapy drugs used in the participants ranged from moderate to highly emetogenic, such as those based on doxorubicin and anthracycline [23,24,25,27]. The chemotherapy drugs used in one study were not reported [26]. 

In three studies, ginger was used as an intervention in the form of capsules containing powdered ginger root; in one study, ginger powder was mixed with yogurt; and in one study, fresh ginger was sliced into 3 cm diameter and 0.2 cm thickness slices. The daily dose of ginger ranged from 0.5 g to 1.5 g, and the treatment period ranged from 3 to 6 days. Participants in the experiment group consumed ginger from three days before to thirty minutes after finishing chemotherapy. Four of the five studies’ participants received appropriate anti-vomiting treatments for both the ginger and control groups [23,24,25,27]. In one study, all subjects received 5-HT_3_ receptor antagonists palonosetron, dexamethasone, an antihistamine, ranitidine, and aprepitant, before the administration of chemotherapy [24]. Three studies used 5-HT_3_ receptor antagonist and/or dexamethasone to manage CINV [25,26,27].

### 2.3. Risk of Bias Associated with Included Trials

The risk of bias ranges from low to high, and Figure 2 shows a summary of the individual studies. One study had a low risk of bias, three studies had a high risk of bias, and one study’s risk of bias was unclear. In all studies, random sequence generation was adequate, but allocation concealment was deemed high risk in one study [24] and unclear in another [27]. In two studies [24,25], performance and detection bias were deemed high risk, while one study [27] was deemed unclear. In one study [23], attrition and reporting domains were rated as high risks. All studies were deemed to be free of other bias.

### 2.4. Effects of Ginger Intake in Managing CINV on Breast Cancer Patients

A total of four studies were included in this meta-analysis. One study was excluded from quantitative analysis because it was combined with ginger intake and acupoint therapy [27]. The merged data revealed that intake of ginger was more effective with a small effect size than control in reducing the severity of CINV in breast cancer patients (SMD −0.32, 95% CI −0.59, −0.05, *p* = 0.02; I^2^ = 75%) [23,24,25,26]. However, because of heterogeneity between samples, we conducted a subgroup analysis and divided nausea, vomiting, and retching into acute or delayed.

#### 2.4.1. Effects of Ginger Intake in Managing Acute CINV on Breast Cancer Patients

Three studies found that ginger consumption was significantly more effective than a control group in reducing the severity of acute CINV in breast cancer patients (SMD −0.48, 95% CI −0.84, −0.13, *p* = 0.008; I^2^ = 66%) [24,25,26]. Figure 3: A subgroup analysis of two studies found that ginger consumption was associated with a significant reduction in acute vomiting severity (SMD −0.56, 95% CI −0.89, −0.22, *p* = 0.001; I^2^ = 0%) [24,26]. One trial (n = 60) found that ginger consumption reduced acute nausea severity more than antiemetic treatment alone (SMD −1.59, 95% CI −2.36, −0.82, *p <* 0.0001) [24].

#### 2.4.2. Effects of Ginger Intake in Managing Delayed CINV on Breast Cancer Patients

Although there was significant heterogeneity [24,25,26], the results showed a significant reduction in delayed CINV in the ginger group compared to the control group (SMD −0.48, 95%CI −0.82, −0.14, *p* = 0.006; I^2^ = 81%). (Figure 4). A pooled analysis of two studies found that the experimental group that took ginger was more effective than the control group in reducing delayed vomiting severity (SMD −0.82, 95%CI −1.36, −0.29, *p* = 0.002; I2 = 77%) [24,26]. One study found that ginger treatment, when compared to antiemetic treatment, significantly reduced the severity of delayed nausea in breast cancer patients (SMD −1.41, 95% CI −1.98, −0.84, *p* < 0.00001) [24]. Subgroup analysis on delayed retching revealed that ginger intake was more effective than antiemetic treatment alone in breast cancer patients (SMD −0.34, 95%CI −0.63, −0.04, *p* = 0.02; I^2^ = 0%) [24].

### 2.5. Safety

In two studies [23,24], no patients reported adverse effects from ginger consumption. Panahi (2012) reported that taking ginger can cause heartburn, headaches, and vertigo. The only side effect reported by Yekta (2012) during the ginger intervention was heartburn. However, there was no statistically significant difference between the ginger and control groups (Table 1).

## 3. Discussion

Nausea and vomiting caused by chemotherapy are both psychologically and physically distressing symptoms. Different treatment regimens are required for acute, delayed, anticipatory, breakthrough, and refractory CINV, which frequently include 5-HT_3_ receptor antagonists, NK1 receptor antagonists, and corticosteroids [2,28]. Despite significant antiemetic agent research and development, CINV management remains a significant challenge, with many unmet needs, such as controlling non-acute CINV, developing appropriate CINV treatment protocols for multiple-day chemotherapy patients, and providing options for those who are prone to CINV despite treatment [28]. Furthermore, common antiemetic drug side effects include headache, constipation, and fatigue [29].

As a result, there is an ongoing demand for complementary and alternative therapies. One of the most promising and actively researched options is herbal medicine. Ginger has been used to treat nausea and vomiting for over 2000 years [30]. Several clinical trials [16,24,31,32] have shown that ginger has an antiemetic effect against both the acute and delayed phases of CINV. Mechanisms of how ginger’s components affect CINV are being researched. The main pungent constituents and fractions of ginger are 6-,8-,10-gingerol and 6-,8-,10-shogaol [33]. These components inhibit 5-HT_3_ receptors in the central and peripheral nervous systems [34,35]. These receptors play an important role in the regulation of peristalsis, pain transmission, and nausea and vomiting. The development of 5-HT3 receptor antagonists improved the treatment of CINV in cancer patients significantly [34].

Various methods may be used to improve ginger accessibility for CINV patients. Ginger partitioned moxibustion may be more effective than no treatment in reducing the severity and frequency of CINV (RR: 2.04, 95% CI: 1.42–2.93); moxibustion may be more effective than antiemetic drugs (RR: 1.87, 95% CI: 1.27–2.76) [35]. Another study found that ginger moxibustion combined with acupuncture reduced gastro-intestinal tract reactions to chemotherapy in cancer patients when compared to a control group [36]. Ginger slice consumption and ginger-applied acupoint therapy were combined in Liu et al. (2020). All levels of CINV (mild, moderate, severe, and very severe) were assessed from 0 to 5 days after chemotherapy, and the severity of CINV between the two groups was significantly reduced (*p <* 0.05), implying that ginger could effectively manage CINV in breast cancer when combined with acupoint therapy [27]. Another study found that direct inhalation of ginger aromatherapy was beneficial [37].

Uncontrolled CINV reduces patients’ quality of life (QOL), as well as their physical and social functioning. It can also result in medical complications such as poor nutrition, dehydration, and electrolyte imbalances [38]. It also leads to patients discontinuing potentially beneficial treatment regimens [38]. Uncontrolled CINV in a patient costs an extra USD 1300 per month in direct medical costs, according to one study [39]. However, current CINV treatments do not appear to be as effective as they could be [2]. Despite the existence of separate NCCN, ASOC, and MASCC/ESMO CINV practice guidelines, they share fundamental similarities and a lack of literature findings [6]. As a result, we determined that it was necessary to assemble evidence-based literature findings for CINV management, which could lead to the development of a new clinical practice guideline recommendation.

Although CINV can occur in any cancer patient receiving chemotherapy, we chose to focus on breast cancer patients for a number of reasons. First, ginger has been shown to be effective against breast cancer. Breast cancer is classified as either ER-positive (as in MCF-7 and T47D cell lines) or ER-negative (as in MDA-MB-231, MDA-MB-468, SKBR3 and MDA-MB-453 cell lines) [18]. Using additional biomarkers such as progesterone receptor (PR) and human epidermal growth factor receptor 2 (HER2), breast cancer is further classified as luminal A, luminal B, basal-like, and HER2-positive [18]. Because these distinct subtypes of breast cancer respond differently to treatment, breast cancer is extremely difficult to treat. As a result, the search for complementary therapeutic methods is ongoing. In MDA-MB-231 cells, for example, methanolic extract of ginger inhibited proliferation and colony formation in a dose- and time-dependent manner [40]. In MCF-7 and MDA-MB-231 cells, ginger extract increased Bax levels while decreasing Bcl-2 proteins, NF-B, Bcl-X, Mcl-1, survivin, cyclin D1, and CDK-4. Furthermore, ginger extract inhibited the expression of two important cancer molecular targets, c-Myc and hTERT [41]. Gingerols were discovered to inhibit breast cancer cell proliferation and metastasis. By inhibiting cyclin-dependent kinases and cyclins, 10-gingerol inhibited MDA-MB-231 proliferation, resulting in a G1 phase arrest [42]. Moreover, 10-gingerol also inhibited cancer cell invasion by inhibiting the activation of Akt and p38 (MAPK) [43]. Furthermore, 6-gingerol inhibited MDA-MB-231 cell migration and motility in a concentration-dependent manner, as well as MMP-2 and 9 expression and activity [44]. Shogaols also inhibited breast cancer cell metastasis via a variety of mechanisms, including MMP-9 inhibition of NF-kB activation, invasion of MDA-MB-231 cells, and inhibiting invasion by decreasing levels of c-Src kinase, cortactin, and MT1-MMP, all of which inhibited the growth and sustainability of breast cancer cells [45,46,47]. Furthermore, because the current meta-analysis on ginger’s antiemetic effect on CINV focuses on all types of cancer and there is a lack of breast cancer focused analysis [20,48], we decided to focus on CINV in breast cancer patients for this study based on previous research demonstrating ginger’s anticancer effect in relieving CINV symptoms, as well as its beneficial effect on breast cancer itself, and the fact that breast cancer has the highest cancer prediction rate [1].

This meta-analysis included 337 patients from four randomized controlled trials, and we discovered that ginger was associated with a reduction in CINV (SMD −0.32, 95% CI −0.59, −0.05, *p* = 0.02; I^2^ = 75%). However, there was significant variation. As a result, we decided to conduct subgroup and sensitivity analysis. Subgroup analysis revealed that ginger was effective in reducing the severity of acute CINV in breast cancer patients (SMD −0.48, 95% CI −0.84, −0.13, *p* = 0.008; I^2^ = 66%). Another subgroup analysis of ginger’s efficacy in acute vomiting revealed statistical significance (SMD −0.56, 95% CI −0.89, −0.22, *p* = 0.001; I^2^ = 0%) (Figure 3). Ginger was also statistically significant in treating delayed CINV in breast cancer patients (SMD −0.48, 95%CI −0.82, −0.14, *p* = 0.006; I^2^ = 81%) (Figure 4). There have been no serious adverse effects reported as a result of ginger consumption. Based on our findings, we concluded that using ginger to treat CINV in breast cancer patients may be both effective and safe. However, because there were insufficient studies included in this analysis, evaluating publication bias using funnel plots was difficult. The sensitivity analysis comparing the fixed effect model and the random effect model yielded similar results. To advance the field and better inform clinicians and patients, more methodologically rigorous research on the safety and efficacy of ginger for CINV in breast cancer patients is required. CINV should be evaluated using validated tools for better meta-analysis. It must also be evaluated in terms of other side effects such as anxiety, depression, diarrhea, and quality of life as a result of chemotherapy.

## 4. Limitations

This research has limitations. First, we only included databases in English and Chinese; while ginger studies may appear in Japanese and Korean literatures, they are unlikely to be large in number and thus would not have significantly influenced our findings. Second, the clinical heterogeneity of the studies included in this review reduces the confidence in the results. Furthermore, the efficacy of ginger was assessed using various instruments; however, few publications reported the same results. As a result, it is critical to use validated tools that can be easily compared to other results. Finally, we were unable to assess publication bias due to the small number of included studies. As a result, more clinical trials in this field are required.

## 5. Methods

### 5.1. Objective

The purpose of this systematic and meta-analysis is to evaluate the efficacy and safety of ginger in the symptomatic management of chemotherapy side effects in breast cancer patients. The current systematic review and meta-analysis followed Preferred Reporting Items for Systematic Reviews and Meta-Analysis (PRISMA) and Cochrane guidelines [21]. PROSPERO was used to register the study protocol (registration no. CRD42022344125).

### 5.2. Criteria to Select Articles

This review included randomized controlled trials of ginger for the treatment of CINV in breast cancer patients that were published in both English and Chinese. There were no restrictions on the study’s origin or publication year. Studies involving participants with CINV symptoms in breast cancer patients were considered eligible for inclusion. There were no restrictions on the participants’ geographic, socioeconomic, or ethnic backgrounds. Ginger consumption is one of the interventions to be considered. Both ginger (Zingiber officinale) alone and ginger combined with other herbs are included. The meta-analysis excludes studies in which ginger is used in conjunction with nonpharmacologic therapy such as acupuncture, massage, far infrared, physical therapy, thermotherapy, or magnetic therapy. There have been studies that compare the efficacy of various ginger formula modifications.

### 5.3. Search Strategy, Data Selection, and Data Extraction

PubMed, the Cochrane Library, Embase, CNKI, and Wanfang were searched until June 2022. Following terms were used: CINV[Title/Abstract], Chemotherapy induced nausea and Vomiting[Title/Abstract], Vomiting[Title/Abstract], Nausea[Title/Abstract], Vomiting[MeSH Terms], Nausea[MeSH Terms], 1 OR 2 OR 3 OR 4 OR 5 OR 6, Breast neoplasms[MeSH Terms], Breast cancer[Title/Abstract], Breast neoplasms[Title/Abstract], 8 OR 9 OR 10, ginger[MeSH Terms], ginger[Title/Abstract], zingiber officinale[Title/Abstract], ShengJiang[Title/Abstract], Zingiberis Rhizoma Recens[Title/Abstract], 12 OR 13 OR 14 OR 15 OR 16, 7 AND 11 AND 17 (Table 2). Endnote 20 (Clarivate Analytics, UK) was used to import search results from the original databases. To ensure objectivity, two authors (S.D.K. and E.B.K.) independently assessed each record’s eligibility using a study screening standard operating procedure. To identify additional eligible trials, reference lists from relevant systematic reviews and meta-analyses were reviewed. We began with the titles and worked our way down to the abstracts, excluding non-clinical trial studies and those that did not focus on ginger or breast cancer symptom management. The same two authors then reviewed the full-text article and documented the reason for exclusion, with disagreements resolved by consensus by another two authors (M.X.Y. and H.S.Y.).

Using Excel, two authors (S.D.K. and E.B.K.) independently extracted detailed data from each study (study origin, year of publication, patient demographics, intervention, comparator, outcome and results, setting, AEs, etc.). Disagreements or disputes were settled through discussions with two additional reviewers (M.X.Y. and H.S.Y). The data required for meta-analysis were transferred from the data extraction form to RevMan software (The Cochrane. Collaboration, Oxford, UK) using a double entry method (version 5.4).

### 5.4. Risk of Bias

Two reviewers (S.D.K. and E.B.K.) used the Cochrane Collaboration’s tool to assess the risk of bias in each included study. Disputes were resolved by two additional reviewers through discussions and arbitration (M.X.Y. and H.S.Y.).

### 5.5. Meta-Analysis

We planned a meta-analysis of all studies that provided efficacy data on CINV in breast cancer patients and used a placebo, conventional medicine, ginger + conventional group as a comparator. We planned to pool data on CINV severity, CINV incidence, acute and delayed CINV efficacy, and safety. RevMan 5.4 was used to synthesize and statistically analyze the efficacy data. Because dichotomous data were unsuitable for synthesis, we decided to only use continuous data. Because the tools used to measure the outcome differed, the effect size was reported using the standard mean difference (SMD). Chi-square tests were performed in the forest plot using RevMan 5.4 to investigate statistical heterogeneity, and a *p* value of less than 0.10 was considered significant, according to the Cochrane Handbook [22]. The I^2^ value was calculated to quantify statistical heterogeneity. We used a fixed effects model for meta-analysis if there was no or low heterogeneity among studies (I^2^ = 50%); if statistical heterogeneity was high (I^2^ > 50%), we investigated sources of heterogeneity using subgroup or sensitivity analysis, and we used a random effects model for meta-analysis. The source of the high heterogeneity was identified using subgroup analysis or sensitivity analysis.

## 6. Conclusions

In conclusion, this review discovered that ginger could be a safe option for breast cancer patients who are suffering from CINV and is associated with symptom improvement in breast cancer patients suffering from CINV. To assess the full effects of ginger for CINV, more rigorous clinical trials focused on reliable outcome measurement and long-term effects are required.

## Figures and Tables

**Figure 1 ijms-23-11267-f001:**
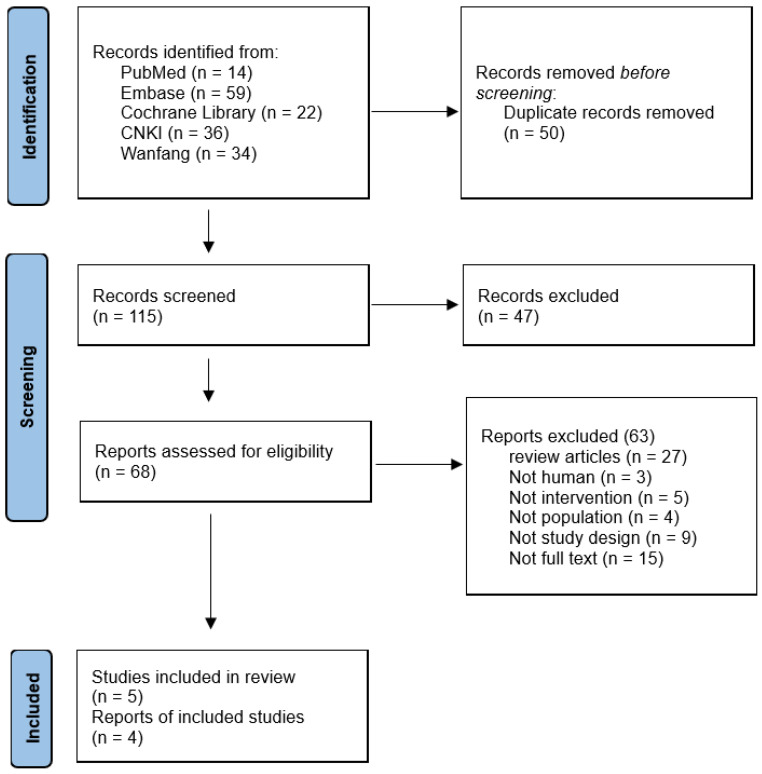
Flow diagram of study selection.

**Figure 2 ijms-23-11267-f002:**
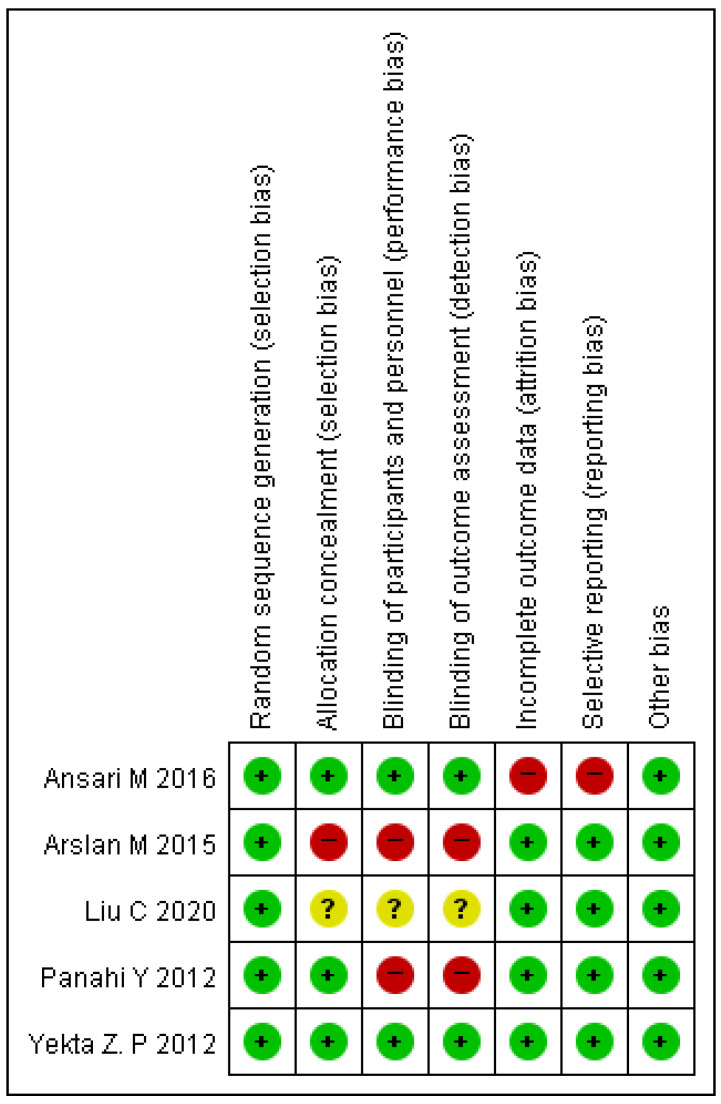
Risk of bias associated with included trials [23,24,25,26,27].

**Figure 3 ijms-23-11267-f003:**
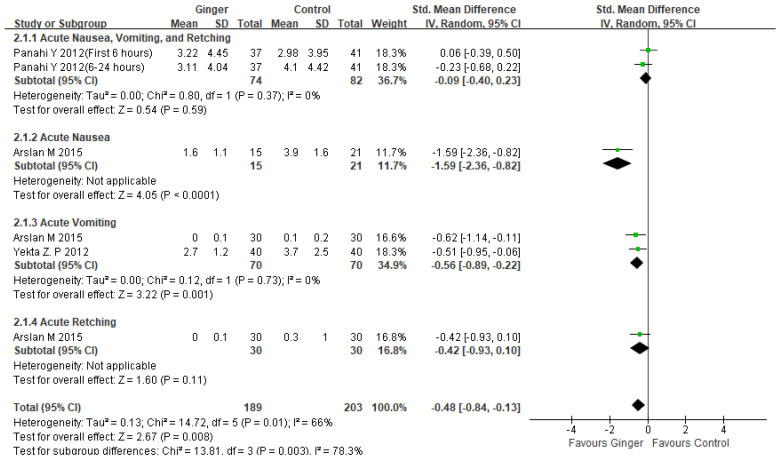
Forest plot of acute CINV severity managements in ginger intake versus controls. CI: confidence interval; SD: standard deviation [24,25,26].

**Figure 4 ijms-23-11267-f004:**
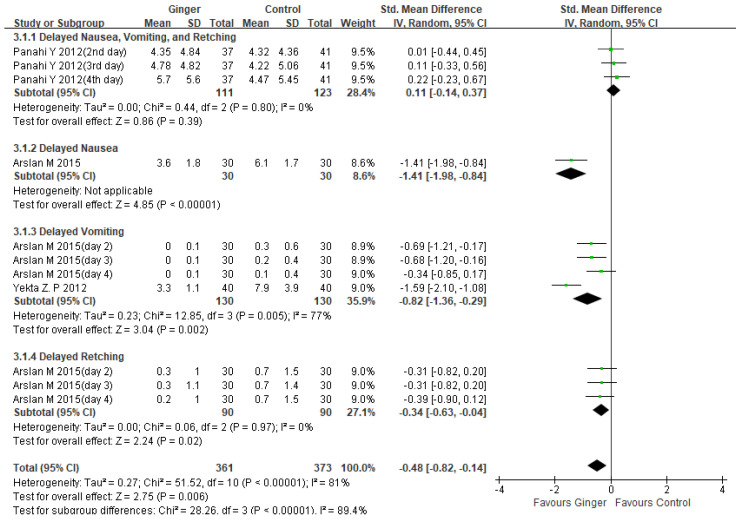
Forest plot of delayed CINV severity managements in ginger intake versus controls. CI: confidence interval; SD: standard deviation [24,25,26].

**Table 1 ijms-23-11267-t001:** Characteristics of included studies.

Study	Study Design	Age	Chemotherapy Drugs	Sample Size	Intervention	Control	Duration	Primary Results	Adverse Effects
Panahi 2012 [25]	Open-label RCT	51.8	Docetaxel, epirubicin, cyclophosphamide	E:37 C:41	Powdered ginger root capsule 1.5 g/d in 3 divided doses every 8 h, antiemetic regimen	Antiemetic regimen (granisetron + dexamethason)	4 days	Ginger was observed no significant advantage on the prevalence or severity of acute or delayed CINV	Heartburn, headache, and vertigo
Yekta 2012 [26]	Double-blind placebo RCT	E:41.8 C:45.1	Not reported	E:40 C:40	Powdered ginger root capsule 1.0 g/d with 6 h, four times a day, antiemetic regimen	Placebo, antiemetic regimen (granisetron + dexamethason)	6 days	Ginger group showed significantly lower vomiting cases of anticipatory, acute, and delayed phases	Heartburn; Anticipatory: E:4, C:1; Acute: E:5, C:0; Delayed E:2, C:0
Arslan 2015 [24]	RCT	48.5	Anthracycline-Based Chemotherapy	E:30 C:30	500 mg powdered ginger, mixed with a yogurt, twice a day, antiemetic	5-HT_3_ antagonist, dexamethasone, antihistamine, ranitidine, aprepitant	3 days	Significantly lower severity of nausea and cases of vomiting in the ginger group than in control group	No adverse effects
Ansari 2016 [23]	RCT	48.6	Doxorubicin-Based Chemotherapy	E:57 C:62	2 Powdered ginger (250 mg per one capsule) capsules every 13 h	Placebo (starch capsule)	3 days	Ginger group exhibited no significant benefits to the severity of nausea and vomiting	No adverse effects
Liu 2020 [27]	RCT	E:51.6 C:52.1	Anthracycline-Based Chemotherapy, cyclophosphamide	E:40 C:40	Slices of fresh ginger under the tongue 30 min before chemotherapy, applied to acupoint, antiemetic	5-HT_3_ antagonist	5 days	Ginger group effectively alleviate chemotherapy-induced nausea and vomiting symptoms	Not reported

**Table 2 ijms-23-11267-t002:** Search strategy.

**PubMed**	(“CINV”[Title/Abstract] OR “chemotherapy induced nausea and vomiting”[Title/Abstract] OR “Vomiting”[MeSH Terms] OR “Nausea”[MeSH Terms] OR “Vomiting”[Title/Abstract] OR “Nausea”[Title/Abstract]) AND (“breast cancer”[Title/Abstract] OR “breast neoplasms”[Title/Abstract] OR “breast neoplasms”[MeSH Terms]) AND (“ginger”[Title/Abstract] OR “ginger”[MeSH Terms] OR “zingiber officinale”[Title/Abstract] OR “ShengJiang”[Title/Abstract] OR “Zingiberis Rhizoma Recens”[Title/Abstract])
**Embase**	(‘breast neoplasms’/exp OR ‘breast neoplasms’:ti,ab,kw OR ‘breast cancer’:ti,ab,kw) AND (‘cinv’/exp OR ‘cinv’:ti,ab,kw OR ‘chemotherapy induced nausea and vomiting’:ti,ab,kw OR ‘vomiting’/exp OR ‘nausea’/exp OR ‘vomiting’:ti,ab,kw OR ‘nausea’:ti,ab,kw) AND (‘ginger’/exp OR ‘ginger’:ti,ab,kw OR ‘zingiber officinale’:ti,ab,kw OR ‘ShengJiang’:ti,ab,kw OR ‘Zingiberis Rhizoma Recens’:ti,ab,kw)
**Cochrane Library**	ID→Search→Hits #1→(“CINV”):ti,ab,kw OR (chemotherapy induced nausea and vomiting):ti,ab,kw OR (Vomiting):ti,ab,kw OR (Nausea):ti,ab,kw→58,816 #2→MeSH descriptor: [Nausea] explode all trees→5992 #3→MeSH descriptor: [Vomiting] explode all trees→5733 #4→#1 OR #2 OR #3→58,851 #5→(breast cancer):ti,ab,kw OR (breast neoplasms):ti,ab,kw→40,959 #6→MeSH descriptor: [Breast Neoplasms] explode all trees→14,567 #7→#5 OR #6→40,961 #8→(ginger):ti,ab,kw OR (zingiber officinale):ti,ab,kw OR (ShengJiang):ti,ab,kw OR (Zingiberis Rhizoma Recens):ti,ab,kw→1075 #9→MeSH descriptor: [Ginger] explode all trees→159 #10→#8 OR #9→1075 #11→#4 AND #7 Additionally, #10→22

## Data Availability

The authors confirm that the data supporting the findings of this study are available within the article.

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
