# Peer review of "Efficacy and Safety of Ginger on the Side Effects of Chemotherapy in Breast Cancer Patients: Systematic Review and Meta-Analysis"

_ijms, 2022, doi:10.3390/ijms231911267_

Round 1
Reviewer 1 Report
Overall a good review. No significant edits suggested from this reviewer
Author Response
Dear, Reviwer.
Thank you so much for your opinion.
We have attached a revised version of the paper that was edited based on the feedback from all reviewers.
Please see the attached document.
Sincerely, Soodam Kim, Eunbin Kwag, Mingxiao Yang, Hwaseung Yoo.

Reviewer 2 Report
This manuscript systematically reviewed and meta-analyzed the efficacy and safety of ginger for the treatment of chemotherapy-induced nausea and vomiting in breast cancer patients based on the published results from 4 papers on clinical trials.
It is written in excellent shape and perfect language. However, considering the high scientific merit requirements of the International Journal of Molecular Sciences, I don’t think it is suitable to be published in this journal, compared with the quality of a review published recently on IJMS (2022, 23, 8812. https://doi.org/10.3390/ijms23158812).
The manuscript has 12 pages, including larger figures, and cited 26 references, which seems inadequate for a review. It would be better If the uses of ginger in other types of cancers were added. Readers would wonder if chemotherapy-induced nausea and vomiting is the side effects only for breast cancer patients or if ginger doesn’t work for treating the CINV in other cancer patients.
Do the authors consider how the chemical constituents (or bioactive compounds) of gingers act on the CINV? Does the source of gingers impact the results of the clinical trials? Do the authors have any ideas to improve the clinical use of ginger for CINV?
In addition, more databases should be surveyed, including some non-English languages. These results don’t have to be analyzed but at least shown. I don’t think it is an appropriate way to say, “these studies in non-English languages are unlikely to be of high quality” (line 226 in this manuscript). This is my personal opinion.
Author Response
Response to the reviewers.
Dear, Reviewer.
Thank you very much for your input. They were both constructive and beneficial. We agree that the number of articles available for meta-analysis is limited. However, we were only able to include articles that met the criteria for inclusion. We have added more explanations for why we chose the inclusion criteria that we did. Furthermore, we searched two Chinese search engines and added one Chinese article that met our criteria.
It was also explained how ginger's bioactive works in CINV pathways. Finally, as you suggested, we investigated various methods of ginger application for CINV in cancer patients, such as moxibustion, aromatherapy, and so on, in order to improve its clinical usage.
Please see the attached document (revised version of the paper).
Again, thank you for the valuable reviews and we will look forward to hearing from you again.
Sincerely, Soodam Kim, Eunbin Kwag, Mingxiao Yang, Hwaseung Yoo.
2022.09.16

Round 2
Reviewer 2 Report
The are a lot of typos, particularly in using the. The following are just several examples.
Since the current treatment option for CINV has several flaws, alternative treatment options are required
Overall, the authors found that ginger was associated with a reduction in CINV.
Ginger's bioactive have has been studied to see how they affect the CINV mechanism.
After duplicates removal (n = 50), we screened….
However, there were was substantial heterogeneity.
and so on.
Author Response
Dear, Reviwer.
Thank you so much for your suggestions.
We went over the script again and corrected the typos and grammar errors. Thank you for addressing the errors. We are extremely grateful.
The updated documentation is attached.
Thank you very much,
Soodam Kim, Eunbin Kwag, Ming Xiao, and Hwaseung Yoo.

Round 3
Reviewer 2 Report
It looks much better.